# Enabling Fast Differentially Private SGD
# via Just-in-Time Compilation and Vectorization

**Pranav Subramani**[*]
Cheriton School of Computer Science
University of Waterloo
`pranav.subramani@uwaterloo.ca`

**Nicholas Vadivelu**[*]
Cheriton School of Computer Science
University of Waterloo
`nbvadive@uwaterloo.ca`

**Gautam Kamath**
Cheriton School of Computer Science
University of Waterloo
`g@csail.mit.edu`

## Abstract

A common pain point in differentially private machine learning is the significant runtime overhead incurred when executing Differentially Private Stochastic Gradient Descent (DPSGD), which may be as large as two orders of magnitude. We thoroughly demonstrate that by exploiting powerful language primitives, including vectorization, just-in-time compilation, and static graph optimization, one can dramatically reduce these overheads, in many cases nearly matching the best non-private running times. These gains are realized in two frameworks: one is JAX, which provides rich support for these primitives through the XLA compiler. We also rebuild core parts of TensorFlow Privacy, integrating more effective vectorization as well as XLA compilation, granting significant memory and runtime improvements over previous release versions. Our proposed approaches allow us to achieve up to 50x speedups compared to the best alternatives. Our code is available at `https://github.com/TheSalon/fast-dpsgd`.

## 1 Introduction

Machine learning has recently experienced tremedous growth, being used to solve problems with unprecedented accuracy in a myriad of domains. However, not all domains are alike—while many datasets are freely available, we often would like to train ML models on sensitive data. Troublingly, it has been demonstrated that disregarding these concerns, or even using heuristic and best-effort privacy approaches, can result in significant leakage of private information [9, 12]. Differential privacy (DP) [16] has emerged as a strong and rigorous notion of data privacy, capable of protecting against privacy violations in a variety of settings.

One of the workhorse algorithms in machine learning is stochastic gradient descent (SGD) which has a differentially private analogue, DPSGD [43, 8, 2] which was introduced as a drop-in replacement for SGD. The primary differences include a per-example gradient clipping step and a batch-level addition of Gaussian noise. While these modifications seem relatively innocuous, they have so far led to non-trivial costs in terms of running time and final accuracy. In this paper, we address and mitigate the running time overhead of DPSGD.

Most modern machine learning frameworks allow efficient access to average minibatch gradients, and not at a per-example level. Access to these objects is critical in DPSGD, as well as other applications

---

[*]Equal contribution.

35th Conference on Neural Information Processing Systems (NeurIPS 2021).

beyond privacy [52]. Lack of support for fast computation of per-example gradients has been noted and lamented numerous times for both TensorFlow [35, 42, 3] and PyTorch [32, 6].

Numerous attempts to avoid these computational roadblocks have been proposed. Goodfellow [20] proposed an algorithmic solution for computing per-example $\ell_2$-norms of the gradients for fully-connected networks. Other proposed solutions work by exploiting Jacobians [14] or parallelizing over the batch dimension [4]. Several of these approaches are are restricted to specific types of architectures—for example, [20] is restricted to fully-connected layers, though [41] extends this to convolutional layers, and a very recent work [33] further considers layers including recurrent networks, attention, and more. BackPACK [14] currently supports only fully connected and convolutional layers, and while the paper states that it can be extended to recurrent and residual layers, GitHub issues related to implementation of these features have been open since November 2019 [13]. Facebook's Opacus [51] emphasizes speed and scalability as the main selling points. We briefly mention microbatching, in which comparatively small subsets of the minibatch called "microbatches" of points are averaged before clipping, reducing the number of clipping operations (and thus the running time), at the cost of requiring additional noise to achieve the same privacy guarantee. Since this generally results in significantly worse accuracy, we do not investigate it further in our work. A more thorough description of approaches is provided in Section 2.1.

As mentioned in the literature (and thoroughly explored later in this paper), all existing approaches seem to incur moderate to severe running time overhead versus non-private SGD, with slowdowns as large as two orders of magnitude. For instance, Carlini et al. [12] comment "Training a differentially private algorithm is known to be slower than standard training; our implementation of this algorithm is 10-100x slower than standard training," where their implementation is based on TensorFlow Privacy. Additionally, Thomas et al. [45] document a slowdown from 12 minutes to 14 hours due to the introduction of differential privacy, a 70x slowdown. The effect of these slowdowns can range from an inconvenience when it comes to rapid prototyping of smaller models, to prohibitively expensive for a single training run of a larger model. Overcoming this obstacle is an important step in helping differentially private machine learning transition from its present nascent state to widespread adoption.

## 1.1 Results

We demonstrate that one can mostly eliminate the significant running time overhead of differentially private SGD by exploiting language primitives such as vectorization, just-in-time (JIT) compilation, and static graph optimization. These features are core primitives within JAX [19, 10] and Tensor-Flow 2 (TF2) [1], both tensor-processing libraries from Google. These frameworks combine JIT compilation backed by the Accelerated Linear Algebra (XLA) [22] just-in-time compiler (JIT) with auto differentiation for high-performance machine learning. As we will see, JAX is consistently the fastest method for running DPSGD, with running times comparable to the non-private case. Our custom TensorFlow Privacy (TFP) implementation (referred to as Custom TFP), which leverages vectorization and XLA compilation in TensorFlow 2, demonstrates similar performance to JAX and significantly outperforms the existing TFP library. These changes have since been merged into TensorFlow Privacy.

Our primary contributions are as follows:

1. We thoroughly benchmark several frameworks and libraries for DPSGD.

2. We extend TensorFlow Privacy to support TF2, more efficient vectorization, and XLA compilation, significantly improving its running time in most cases (referred to as Custom TFP in this paper). We also contribute a variant of our implementation to TensorFlow Privacy, which is now the fastest DPSGD algorithm the library provides.

3. We demonstrate that methods which use vectorization, JIT compilation, and static graph optimization are consistently the fastest and most memory-efficient: specifically, JAX and Custom TFP.

4. We find that, despite similarities in the compilation pipeline, JAX is generally faster than Custom TFP. We examine and discuss compiled XLA assembly to explain the discrepancy.

5. Finally, our supplement contains code to reproduce these experiments, as well as guide researchers and engineers in developing fast code for private ML.

Table 1: Median running time (s) per epoch of training various models at batch size 128. FCNN stands for Fully-Connected Neural Network; CNN stands for Convolutinal Neural Network.

| Architecture | Private Training | | Non-Private Training |
| | JAX/Custom TFP | Best Alternative | Best Time |
| --- | --- | --- | --- |
| FCNN | **0.21** | 0.77 | 0.55 |
| CNN | **7.3** | 12 | 1.7 |
| LSTM | **8.2** | 407 | 4.8 |

Table 1 summarizes some of our experimental results, with median running time per epoch for a variety of settings. JAX and Custom TFP are consistently the fastest.

We observe dramatic improvements for LSTMs [28], potentially significant enough to bring LSTMs from impractical into the realm of feasibility. JAX is able to privately train these models 17x and 50x faster than the best alternative.[2] Examining the overhead due to privacy: JAX's running time increases by roughly 2x, compared to factors closer to 10x for alternatives.

For fully-connected and convolutional networks, JAX or Custom TFP almost entirely remove the overhead due to privacy. In fact, the running times are significantly better than some alternatives *without* privacy. Recall that these are per-epoch times: while an improvement of 0.5 seconds might seem insignificant, this can add up when training for many epochs. We perform an ablation study (Table 2) for all models to pinpoint the source of all improvements.

While our investigations show the consistent and substantial superiority of JAX for fast private machine learning, these benefits remain relatively unknown. Though a small number of experts are aware [44], and the official JAX repo contains a toy demonstration [25], before the initial posting of our paper a Google Scholar search revealed only two papers which use JAX for differential privacy [49, 38], and neither emphasizes or even comments on the computational advantages of JAX. Similarly, while efficient per-example gradients have been studied in TensorFlow [5], efficient application of these techniques is not readily available to privacy researchers. We hope that our investigation will document this phenomenon and encourage others to adopt it for their private machine learning needs.

## 1.2 Simultaneous and Subsequent Work

Simultaneous to our work, [11] employ Johnson-Lindenstrauss projections to quickly approximate per-example gradient norms. This is an algorithmic modification, and will not be functionally equivalent to DPSGD – similar to microbatching, there is a time-accuracy tradeoff (though not as severe in this case).

Subsequent to the initial posting of this paper, we worked with Google engineers to implement our improvements into TensorFlow Privacy. Vadivelu contributed an JAX implementation of DPSGD to the Optax library [27]. [7] employed our findings to efficiently privately train BERT-Large. In a recent Opacus whitepaper [51], the authors repeat some of our experiments on more recent versions of these frameworks; we defer to their work for discussion of these results.

## 2 Description of Approaches

### 2.1 Libraries Enabling DPSGD

**JAX [19, 10].** JAX is a recently introduced framework for machine learning, defined by its automatic differentiation capabilities and JIT compilation via the XLA compiler [22]. Programs written in pure Python and JAX's NumPy [26] API can be translated to an intermediate language (XLA-HLO) to be JIT compiled, i.e., to generate custom assembly instructions for the hardware. This enables optimizations such as kernel-fusions, buffer reuse, improved memory layout, and more. Additionally, one of the core functions present in JAX is VMAP, a vectorized map, which enables easy-to-write and efficient batch level parallelism that is fundamental to DPSGD. As we will demonstrate, these enable the fastest approach for DPSGD that we are aware of.

---

[2]Note that a confirmed bug in TF2 currently prevents us from running Custom TFP in these cases.

**Custom TFP.** Vectorization and XLA-driven JIT compilation is also available in TensorFlow 2 [1], which we leverage in our implementation, Custom TFP. With these primitives, we achieve performance comparable to JAX and surpassing existing DPSGD implementations in TensorFlow. We augment TensorFlow Privacy to better utilize `tf.vectorized_map` and follow TensorFlow 2 best practices while retaining the existing functionality.

**Chain-Rule-Based Per-Example Gradients [20, 41].** This suite of techniques is implemented on top of PyTorch [39]. They support efficient GPU-accelerated per-example gradients for fully-connected layers via [20], as well as convolutional layers via [41], which we describe in the detail in the following paragraphs.

Let $C, D, T$, and $B$ refer to the number of input channels, output channels, the spatial dimension, and the batch size. The shape of the input $x$ is $(B, C, T)$. The conventional formula for the discrete convolution can be written as:

$$\sum_{c=0}^{C-1} \sum_{k=0}^{K-1} x[b, c, t + K] h[d, c, k].$$

The gradient of this expression can be efficiently computed via automatic differentiation [40]. PyTorch's automatic differentiation cannot be parallelized across the batch dimension $b$ [41], which is required to backpropagate through the above expression. Instead, they rewrite the convolution as follows:

$$\sum_{c=0}^{C/G-1} \sum_{k=0}^{K-1} x \left[ b, c, g\frac{C}{G}, t + K \right] h[d, g, c, k],$$

where $G$ is the number of groups and the shape of $x$ is $(1, B, C, T)$. The initial convolution is 1-dimensional, while the above expression includes an added dimension. Similarly, to allow back-propagation through a $k$-dimensional convolutional layer, a $(k + 1)$-dimensional convolutional layer is required. This can be achieved by utilizing the `group` attribute in the convolution function in PyTorch, since splitting it into groups implies that the same convolution is applied to each individual group.

**BackPACK [14].** The chain rule gives the following expression for the gradient of a loss function:

$$\nabla_{\theta^{(i)}} \ell(\theta) = (J_{\theta^{(i)}} z_n^{(i)})^T \left( \prod_{j=i}^{L-1} (J_{z_n^{(j)}} z_n^{(j+1)})^T \right) (\nabla_{z_n^{(L)}} \ell_n(\theta)).$$

In order to compute this quantity, one requires the ability to multiply the Jacobian by a vector and by a matrix, which is not currently supported in PyTorch's automatic differentiation framework. In BackPACK, Dangel et al. [14] extend several layers within PyTorch to support fast Jacobian-vector and Jacobian-matrix products in order to extract quantities like individual gradients, variance, $\ell_2$-norm of the gradients, and second-order quantities. In particular, to extract first-order gradients, their method multiplies the transposed Jacobian with the outputs of the layer:

$$\frac{1}{N} \nabla_{\theta^{(i)}} \ell(\theta) = \frac{1}{N} (J_{\theta^{(i)}} z_n^{(i)})^T (\nabla_{z_n^{(i)}} \ell(\theta)),$$

where $i = 1, \ldots, N$ and each $\theta^{(i)}$ has a gradient which is of shape $(N, d^{(i)})$. BackPACK provides efficient computation for the transpose of the Jacobian as well as the Jacobian.

**Opacus [51].** Opacus is a library for training PyTorch models with differential privacy, recently released by Facebook. It supports per-example gradients, using PyTorch's forward and backward hooks to propagate gradients. They provide support for several PyTorch layers including LSTM layers, which are not supported in either of the previous two frameworks. Note that Opacus does not support PyTorch's `nn.LSTM` but instead implements a separate `opacus.layers.DPLSTM`, with adjustments that allow individual gradients to propagate through it.

**PyVacy [48].** Before the release of Opacus (and its predecessor PyTorch-DP), PyVacy was the most popular library for DP machine learning in PyTorch. PyVacy has no custom support for parallelization across the batch dimension for any layer since it processes each sample individually (by way of a for-loop). This generally leads to a large increase in runtime for models trained using PyVacy.

**TensorFlow Privacy [37]**   TensorFlow Privacy is a library for differentially private machine learning, built on top of TensorFlow. TensorFlow Privacy has general support for a vectorized implementation of DPSGD via `vectorized_map` which allows it to parallelize across the batch dimension, used to extract per-example gradients. The library recently introduced a TensorFlow 2 compatible API that leverages `GradientTape.jacobian` to compute per-example gradients, which we compare seperately to the TensorFlow 1 API in our experiments.

## 2.2   Notable Framework Features

**Static versus Dynamic Graph.**   TensorFlow and JAX use a *static graph* to track computation in order to optimize execution and compute gradients. This means the sequence of operations is traced and a large proportion of shapes are determined during the first invocation of the function, allowing for kernel fusion, buffer reuse, and other optimizations on subsequent calls. PyTorch uses a *dynamic graph* to track computation flow in order to compute gradients, but does *not* optimize execution. This enables increased dynamism in the shapes and types of computations, at the cost of losing all the aforementioned optimizations.

**Grappler versus XLA.**   TensorFlow has two optimization engines: Grappler [21] and XLA. Grappler, TensorFlow's original graph optimizer, takes as input the computation graph and is able to prune dead nodes, remove redundant computation, improve memory layouts, and more. XLA, TensorFlow's new optimizing compiler, can perform the same optimizations as Grappler, in addition to generating code for fused kernels. For this reason, XLA has the potential to extract more performance out of TensorFlow graphs than Grappler, but does not always accomplish this due to Grappler's maturity.

**JAX and XLA.**   JAX was built from the ground up to leverage XLA, and so many of its operations map directly to XLA primitives. We often observe that JAX is able to extract better performance out of XLA than TensorFlow.

**Pytorch and Static Graphs.**   Recently, PyTorch has released the capability to JIT compile its code through `torch.jit` or PyTorch XLA [18]. Due to the early nature of these two efforts, they were not successful in JIT compiling the methods we tried, thus we do not consider them further.

**Just-In-Time Compilation**   JIT compilation is a method of compilation that happens at runtime, as opposed to before program execution. JAX and TensorFlow perform JIT compilation by recording operations that are executed on tensors/arrays (i.e. "tracing"), generating the low-level instructions to perform these operations, optimizing these instructions, then producing fast low-level kernels. The XLA compiler requires all array shapes to be known at trace-time so it can statically determine how much memory should be allocated for each operation. While compilation can be relatively slow compared to execution (~10x the time), you only pay this price once at the first training iteration, provided the input shapes do not change.

## 3   Empirical Findings

We evaluate the aforementioned implementations of DPSGD in runtime and memory consumption on three datasets: **CIFAR10** [31], a dataset of small colour images with 60,000 training examples of size $32 \times 32 \times 3$ each, **IMDb** [36], a movie review sentiment classification dataset with 25,000 training examples padded to a sequence length of 256 each, and **Adult** [15], containing 45,220 examples with 104 features, which was preprocessed via methods from [29]. These datasets are available for open use and do not contain personally identifiable information or offensive content.

We perform our evaluations on three different architectures. We start with the smallest dataset, Adult, training a 5,532-parameter fully-connected neural network (FCNN). Then, we train a CIFAR10 convolutional neural network classifier architecture with 605,226 parameters used by Papernot et al. [38]. For IMDb, we use an LSTM network with 1,081,002 parameters, demonstrating the method on a relatively large model. This selection covers the common data and architecture types at realistic sizes for differentially private learning. In particular, we did not consider the exceptionally large models which are not prevalent in non-private machine learning. Additional experiments can be found in the supplement that are omitted due to space constraints. These experiments cover a wider range of parameters and architectures to elucidate the benefits of using XLA JIT for DPSGD.

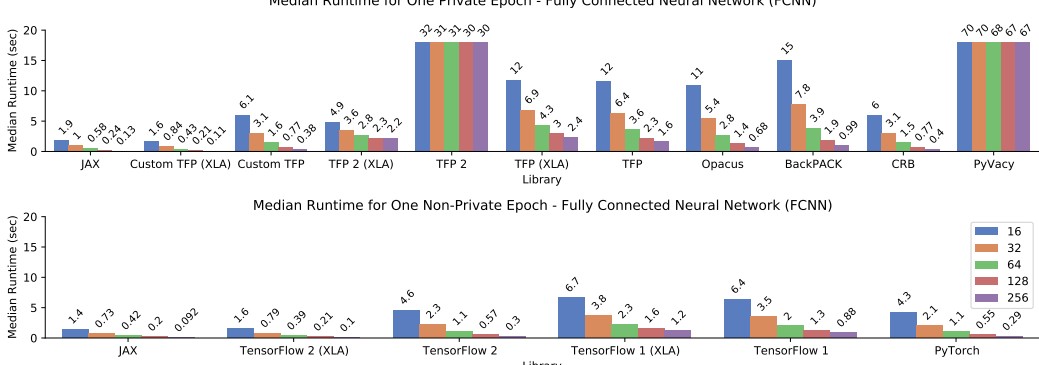

Figure 1: **Runtimes for the fully connected network on the Adult dataset**. We observe that JAX and Custom TFP are the fastest by a large margin in both settings, with DPSGD having low overhead over the non-private setting. The y-axis is truncated for clarity.

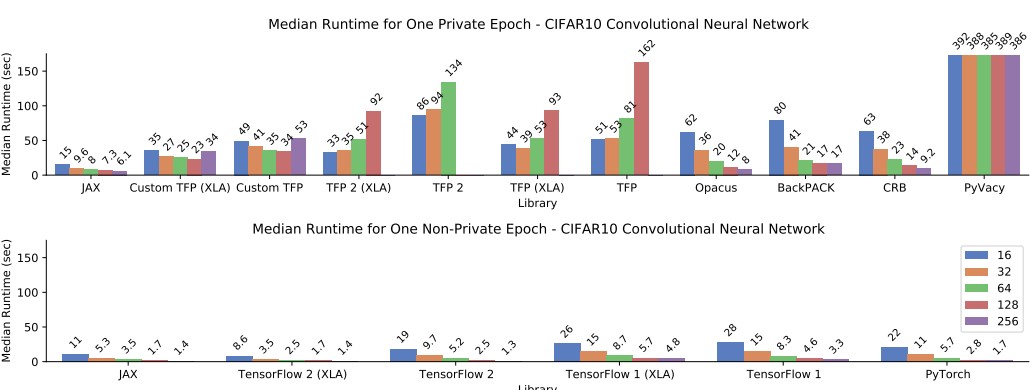

Figure 2: **Runtimes for the CNN on the CIFAR10 dataset.** We observe that JAX has the fastest runtime in the private case while TensorFlow 2 with XLA and JAX are the fastest in the non-private case at most batch sizes. Similar to the MNIST case, TFP struggles at the largest batch size due to an inability to properly parallelize per-gradient computation with this level of memory consumption. The y-axis is truncated for clarity.

We compare a number of different methods for fast DPSGD, including JAX [19, 10], BackPACK [14], the chain-rule based method (CRB) [20, 41], Opacus [51], PyVacy [48], TensorFlow Privacy (TFP) [37] and our modification of TFP, dubbed Custom TFP. For TensorFlow based frameworks, we evaluate performance both with and without XLA JIT compilation in both TF 1 and TF 2. We refer to TensorFlow 2 and JAX as *modern XLA-compiled libraries*.

These architectures and datasets are evaluated in terms of runtime at batch sizes 16, 32, 64, 128, and 256. This showcases a comparison of runtimes across a variety of batch sizes to present a holistic picture of the running times, as well as demonstrating the impact of memory utilization on runtime. Each experiment was run for 20 epochs and the median epoch run-time is reported. The variance of these experiments was low enough that the errors bars are negligible (full results provided in the supplement). Outside of the initial compilation time required for the static graph frameworks, the runtime showed little variance between epochs, resulting in narrow confidence intervals for the epoch runtime (data available upon request). We preprocess all the data in advance and use an identical generator-based dataloader for all frameworks, to ensure consistency.

All experiments were run on Ubuntu 18.04 with an Intel Core i7-7800X CPU (3.50 GHz, 6 cores), NVIDIA GTX Titan V GPU (12GB VRAM), and 32GB of RAM. The code is provided in the supplement and is publicly released.

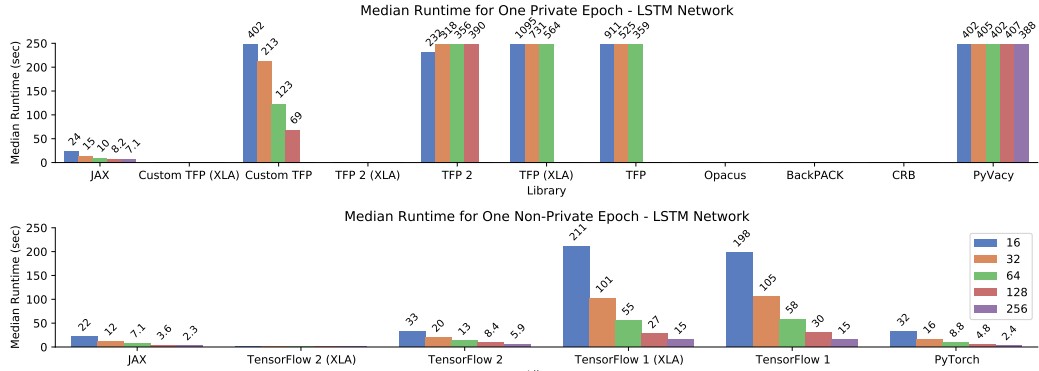

Figure 3: **Runtimes for the LSTM network on the IMDb dataset.** JAX is by far the fastest option, resulting in a roughly 50x speedup for batch size 256. The quadratic memory cost of Opacus prevents us from evaluating this implementation at these batch sizes, however, we observe a median runtime of 1024.16s at batch size 10. Excessive memory consumption prevent us from evaluating Custom TFP and TFP at larger batch sizes. An open TensorFlow 2 bug prevents us from evaluating Custom TFP (XLA) in this setting [46]. BackPACK and CRB do not support embedding layers. The y-axis is truncated for clarity.

First, we evaluate the FCNN model (Figure 1), where we observe that JAX and Custom TFP are significantly faster than the other options in both the private and non-private setting. With such a simple architecture, the compiler can perform significant optimizations. Notably, JAX and Custom TFP show little overhead over their non-private counterparts. The non-statically compiled frameworks (apart from PyVacy) remain competitive in this setting due to the shallow network size and low parameter count. TFP 2 without XLA performs unexpectedly poorly, due to the Jacobian computation which does not account for per-example independence of gradients.

We then evaluate the CIFAR10 CNN model (Figure 2), where JAX is by far the most performant implementation in the private case, and the modern XLA-compiled frameworks are the fastest non-private methods. Custom TFP is noticeably slower than JAX—we conjecture that this is due to different utilization of the JIT compiler, see Section 4. Also, at the largest batch size, TFP's performance deteriorates due to the memory consumption preventing it from effectively parallelizing the per-example gradient computation.

For the final runtime experiment we evaluate the LSTM network (Figure 3). A TensorFlow 2 bug [46] and lack of support from BackPACK and CRB prevent us from evaluating this setting on those implementations. We see a similar phenomenon to the CIFAR10 CNN case for TFP: at larger batch sizes, the library fails to parallelize effectively pessimizing the runtime. TensorFlow and PyTorch benefit from a fast cuDNN LSTM implementation in the non-private case which they fail to leverage in the private case, explaining the significant difference in performance. JAX, on the other hand, uses an LSTM implementation based on primitive operations, which allows it to retain similar performance in both the private and non-private settings.

To understand the importance of vectorization (via `VMAP`) and JIT compilation (`JIT`), we ablate JAX's performance on these tasks with and without these two components (Table 2). We observe that `JIT` alone provides up to a 435x improvement, and `VMAP` alone provides up to a 64x improvement. When used in tandem, both complement each other, providing up to a 5160x improvement.

Similarly, we ablate the components in Custom TFP (Table 2). In TensorFlow, `vectorized_map` automatically compiles the code, preventing us from ablating it alone. We observe that XLA in TensorFlow 2 is able to reclaim performance lost by not vectorizing, seeing that the non-VMAP XLA performance comes close to VMAP Graph performance in many settings. We further see the non-compiled runtimes are not as extreme as seen in JAX: TensorFlow is better optimized to run reasonably fast in all settings.

Finally, we explore the memory consumption behaviour of these implementations (Table 3), observing that running time has a strong negative correlation to memory consumption. The modern XLA-

Table 2: **Ablation of JIT compilation and vectorization in Custom TFP (left) and JAX (right).**
Median runtime per epoch for a run of 20 epochs at batch size 128 for DPSGD. In TensorFlow, one can not use vectorization without graph compilation, which is why there is no standalone vectorization. Empty entries are due to confirmed active bugs in TensorFlow [47, 46]. We exclude LSTMs, as JAX runs out of memory without JIT compilation for this model.

| Graph | JIT | VMAP | FCNN | CNN | LSTM |
|---|---|---|---|---|---|
|  |  |  | 147 | 561 | 717 |
| ✓ |  |  | 12.7 | 85.5 | 361 |
|  | ✓ |  | 1.83 | 71.5 |  |
| ✓ |  | ✓ | 0.770 | 33.8 | 68.6 |
|  | ✓ | ✓ | **0.209** | **23.3** |  |

| JIT | VMAP | FCNN | CNN |
|---|---|---|---|
|  |  | 1240 | 3840 |
|  | ✓ | 19.2 | 64.8 |
| ✓ |  | 2.85 | 84.0 |
| ✓ | ✓ | **0.239** | **7.28** |

Table 3: **Maximum batch size supported by each library before encountering out of memory errors.** Missing entries represent missing functionality or bugs in the frameworks. PyVacy handles examples sequentially, giving constant memory consumption with respect to batch size.

| Library | CNN | LSTM |
|---|---|---|
| JAX | 10,448 | **11,984** |
| TensorFlow 2 (XLA) | **15,040** |  |
|  |  |  |
| TensorFlow 2 | 11,328 | 9,221 |
|  |  |  |
| TensorFlow 1 (XLA) | 10,880 | 5,070 |
| TensorFlow 1 | 11,480 | 5,264 |
| PyTorch | 10,752 | 9,943 |

| Library | CNN | LSTM |
|---|---|---|
| JAX (DP) | **4,264** | **2,487** |
| Custom TFP (XLA) | 3,144 |  |
| TFP 2 (XLA) | 168 |  |
| Custom TFP | 1,944 | 137 |
| TFP 2 | 104 | 105 |
| TFP (XLA) | 168 | 88 |
| TFP | 104 | 105 |
| Opacus | 1,920 | 10 |
| BackPACK | 1,216 |  |
| CRB | 2,184 |  |
| PyVacy | $\infty$ | $\infty$ |

compiled libraries provide impressive batch size capability. Also, all the frameworks except JAX and PyVacy struggle with batch sizes on the LSTM. Since PyVacy processes examples sequentially, it has a constant memory consumption with respect to batch size, effectively trading off running time for optimal memory use. The other frameworks are crippled without access to their fused cuDNN LSTM implementation, while JAX has no issues as its LSTM is composed of primitives. Finally, due to the specialized per-example gradient computation afforded by CRB for convolutions, it shows the best memory utilization among the PyTorch frameworks, even beating Custom TFP (without XLA) in the CIFAR10 CNN case.

## 4  Discussion

JAX and Custom TFP's runtime advancements can be primarily attributed to the advancement of the compiler present in both of these languages. The XLA compiler performs a variety of operations ranging from memory scheduling to kernel fusion. The memory optimizations are vital for larger models where DPSGD becomes a memory-bound algorithm. One of the core features of XLA is buffer reutilization which has a significant impact on the maximum memory used [22]. Furthermore, the memory scheduler can mitigate peak memory usage to prevent a runtime exception for overusing available memory.

The effectiveness of XLA is demonstrated through the peak batch size experiment (which serves as a proxy for memory efficiency): in both the private and non-private settings, XLA far exceeds alternatives in the peak batch size it supports. Through this experiment, we also see the benefits of using small operation primitives as opposed to large fused kernels: TensorFlow and PyTorch both leverage the optimized cuDNN kernel in the non-private setting for performance [23, 17], but cannot in the private setting, leading to significantly worse performance. Modifying all existing cuDNN kernels to enable use in the private case would require a non-trivial engineering investment. JAX instead focuses on optimizing operation primitives, so even in foreign computational circumstances,

its performance is comparatively strong. Succinctly, JAX sacrifices the ability to use highly optimized fused kernels for generalizability.

Our implementation of Custom TFP leverages the vectorized map for both the forward and backward pass, unlike existing implementations which only use vectorized map for the backward pass (identical to the JAX version). This gives the compiler explicit information about the independence of batches in the computation, enabling significant optimizations, explaining the improvement in Custom TFP compared to the existing TensorFlow implementations. Details about the implementation are provided in the supplementary code.

We notice that the runtimes and memory consumption are different between Custom TFP and JAX despite having the same backend compiler. We investigate differences in the XLA-assembly for simple code segments in both frameworks in the supplemental material. TensorFlow 1.0 does not have the same capability to integrate with XLA as TF 2.0 does. The primary optimization available is autoclustering, which we observe does not optimize operations nearly as much as the full JIT.

Through the ablation study of `VMAP` and `JIT` in JAX shown in Table 2, we observe that both components complement each other. In general, we observe that JIT provides the larger performance gain, as shown with the FCNN. For the CNN, the large matrix operations coupled with JAX's asynchronous execution [34] allow reasonable utilization of the GPU even without `VMAP`, which is why we observe less of an improvement from `JIT` alone in this experiment. The runtimes without these two primitives is significantly slower than the other frameworks–this is because JAX was built ground-up to leverage these primitives [10].

In the ablation for Custom TFP in Table 2, we see some key differences with JAX. First, the mechanism for `VMAP` in TensorFlow 2 is different from that in JAX: JAX performs op-by-op batching without compilation, while TensorFlow always compiles the vectorized map [24]. We observe that the non-compiled code still runs in a reasonable amount of time since TensorFlow is optimized to have a competitive eager execution, while JAX is not. Also, in TensorFlow, XLA's JIT compilation without vectorization is often able to bring the runtime performance close to that of the graph mode with vectorization, implying that XLA is able to recognize opportunities for parallelization even when the user does not explicitly request it. Finally, we observe the benefit of having a fast, fused implementation for LSTMs: while JAX ran out of memory outside of a compiled and vectorized context, Custom TFP is able to achieve reasonable runtimes by leveraging the fused kernel.

### 4.1 Drawbacks

While integrating XLA into DPSGD presents a massive runtime and memory advantage, there is a cost. XLA is a subset of all permissable operations in JAX and TensorFlow, requiring users to be cognizant of functionality they use. For example, `jax.numpy.unique(x)` produces a result whose shape is not known at compile time, preventing it from being used with JIT compilation. More subtle errors involving unintended recompilation of code are also possible which can lead to enormous slowdowns.

Another potential downside is that these benefits are not observed if the batch size is large enough and the bottleneck becomes the actual network evaluation. In a situation like this, all of the frameworks would perform at approximately the same speeds. However, in practice, this is rarely true and moreso in differentially private machine learning where models are not particularly large.

## 5   Conclusion and Future Work

We have demonstrated that language primitives like vectorization, JIT compilation, and static graph optimization can dramatically improve the running time of DPSGD, realized by JAX and our Custom TFP. In particular, we find that using JAX can almost entirely remove the computational overhead introduced by DPSGD, thus alleviating a major pain point of private machine learning practitioners.

In our work, we focus on conventional set-ups for academic researchers; for future work, it would be insightful to explore the performance of distributed DPSGD, as distributed set-ups are becoming increasingly commonplace. Furthermore, implementing a PyTorch `JIT` compatible version of DPSGD could provide an alternative to TensorFlow and JAX, particularly if said implementation is compatible with PyTorch XLA. Though these two compilation systems are immature compared to TensorFlow

and JAX, they are rapidly improving and should not be ignored. Outside of Python, there are powerful autodifferentiation methods in other more perfomant languages such as Julia [30] and Swift [50] which are worthy of study.

## Acknowledgements

We would like to thank Roy Frostig for helpful discussions on JAX, Steve Chien and Shuang Song for their work in implementing our improvements in TensorFlow Privacy, Xi He and Om Thakkar for valuable feedback on drafts of this work, and several JAX, TensorFlow, and Opacus developers who helped answer our issues, including James Bradbury, Peter Buchlovsky, Peter Hawkins, Matthew Johnson, Karthik Prasad, Github handle ravikyram, and Qianli Scott Zhu.

## Funding Transparency Statement

Funding in direct support of this work: an NSERC Discovery Grant, a Compute Canada RRG grant, and a University of Waterloo startup grant.

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
