# OpenReview forum: "Enabling Fast Differentially Private SGD via Just-in-Time Compilation and Vectorization"
_NeurIPS.cc/2021/Conference — NeurIPS 2021 Poster_

### Official Review · Reviewer_27Gt · 2021-07-15

**Rating:** 7
**Confidence:** 4

**Summary:**

This paper investigates three language primitives, namely vectorization, just-in-time (JIT) compilation and static graph optimization, that can be leveraged to speed up the process of stochastic gradient descent (DPSGD) in training ML models with differential privacy. It demonstrates by experiments that these techniques can lead to significant reductions in per epoch training time of neural networks compared to naïve DPSGD. It also performs ablation studies to further investigate the impact of each technique and find that JIT alone provides the most improvements.

**Limitations And Societal Impact:**

The authors adequately addressed the potential negative societal impact of their work by pointing out that improved efficiency of DPSGD could increase the accessibility to differential privacy which, when not used properly, could lead to a misleading or false sense of privacy, and suggesting that these issues can be avoided via training and consultation.

**Main Review:**

This paper aims to address a very important problem: to alleviate the significant and sometimes prohibitive overhead of training ML models with DPSGD. Currently, DPSGD is the most widely adopted general method to train neural network models with privacy protections for user data in the training set. However, its use is limited partly due to the large overhead current implementation of DPSGD will incur. Making DPSGD faster will increase its adoption in academia and industry.

The authors benchmark several frameworks and libraries for DPSGD and demonstrate that vectorization, JIT compilation and static graph optimization leads to faster and more memory efficient DPSGD implementations. They extend TensorFlow Privacy to support TensorFlow 2 and contribute a variant of their implementation to TensorFlow Privacy.

 Strength:
+ The problem this paper aim to address is an important one.
+ The experiments are clearly designed and demonstrate that JIP and vectorization lead to significant improvement in per epoch training time.
+ The paper is overall well written. The presentation is clear.

Weakness:
- The biggest weakness is that the contribution of this paper is unclear. While it is good to have experiments to show that vectorization, JIT and graph optimization can improve the performance of DPSGD, these findings are not surprising. JIT and vectorization are known to provide big improvements in training efficiency. They are supported in existing compilers like JAX and XLA, and in TensorFlow and PyTorch for training (non-private) models. Thus, it seems that the biggest contribution of this work is on the engineering side: the authors contribute a variant of their implementation to TensorFlow Privacy and extend TensorFlow Privacy to support TensorFlow 2. This is not saying that the contribution is not important. Such implementation, if made publicly available, is a valuable addition to the DP and ML community. However, the paper did not provide details for the reviewers to assess those contributions. What are the technical difficulties? How much engineering efforts are required to solve those difficulties? I suggest that the authors revise the manuscript to clearly state the novelty of their work.

- The author did experiments on three neural networks: a fully connect network with ~5.5K parameters, a CNN with 605K parameters and a LSTM with ~1M parameters. While these experiments are nice, I would also like to see if their method scales to very large models frequently used in vision and NLP, like BERT or GPT2. Those large models have hundreds of millions or even billions of parameters. It is important for privacy practitioners to know the feasibility to train those big ML models, or just to fine tune them with a downstream task using fast DPSDG.

Other Comments
- The author mentioned in line 258 that “…TensorFlow and PyTorch both leverage the optimized cuDNN kernel in the non-private setting for performance, but cannot in the private setting, leading to significantly worse performance.” However, the authors did not explain why cuDNN cannot be used in the private setting. Is it just an engineering issue or not feasible at all?
- The author mentioned in their contributions (line 77): ”…JAX is generally faster than Custom TFP. We examine and discuss compiled XLA assembly to explain the discrepancy.” However, such discussion is not present in the paper.

Typos
line 35: are are -> are

**Time Spent Reviewing:**

6

---

> ### Author Response · Authors · 2021-08-11
> **Response to reviewer 27Gt**
>
> We thank the reviewer for their careful and thorough reading, and their encouraging and thoughtful comments. We are glad that they find the problem important, the experiments convincing, and the paper well-written.
>
> *Comment*: The biggest weakness is that the contribution of this paper is unclear. While it is good to have experiments to show that vectorization, JIT and graph optimization can improve the performance of DPSGD, these findings are not surprising. JIT and vectorization are known to provide big improvements in training efficiency. They are supported in existing compilers like JAX and XLA, and in TensorFlow and PyTorch for training (non-private) models. Thus, it seems that the biggest contribution of this work is on the engineering side: the authors contribute a variant of their implementation to TensorFlow Privacy and extend TensorFlow Privacy to support TensorFlow 2. This is not saying that the contribution is not important. Such implementation, if made publicly available, is a valuable addition to the DP and ML community. However, the paper did not provide details for the reviewers to assess those contributions. What are the technical difficulties? How much engineering efforts are required to solve those difficulties? I suggest that the authors revise the manuscript to clearly state the novelty of their work.
>
> *Response*: We thank the reviewer for recognizing the importance of engineering work. We will include the details of relevant challenges in the final manuscript, which primarily includes the training pipeline structure due to the nature of the vectorization. We also made this implementation public by contributing it to Optax and TensorFlow Privacy and including the code in the supplement.
>
> We want to emphasize that, while some of our findings may be natural to experts in ML performance engineering, aspects were novel to domain experts in several areas. For instance, our improvements were not previously implemented by the TensorFlow Privacy team, which includes experts familiar with the low level workings of TF, despite slow running time of DPSGD being a significant pain point. Similarly, in correspondence with a developer of JAX, they were impressed by the magnitude of the performance improvement realized in our paper. Finally, another group recently used our techniques to enable efficient fine-tuning of BERT-Large [1]. In all cases, our findings either exhibited surprising behavior to domain experts or enabled functionality which may have previously been intractable. Thus, particularly to the privacy community which our paper targets, we believe this demonstrates both novelty and significance of our work.
>
> *Comment*: The author did experiments on three neural networks: a fully connect network with ~5.5K parameters, a CNN with 605K parameters and a LSTM with ~1M parameters. While these experiments are nice, I would also like to see if their method scales to very large models frequently used in vision and NLP, like BERT or GPT2. Those large models have hundreds of millions or even billions of parameters. It is important for privacy practitioners to know the feasibility to train those big ML models, or just to fine tune them with a downstream task using fast DPSDG.
>
> *Response*:  A very recent paper on arXiv did exactly this investigation: specifically, [1] is inspired by our approach, and privately fine tunes BERT-Large (340M parameters) using JAX. They comment that these experiments were enabled by precisely the features we highlight. From a software engineering perspective, our implementation Custom-TFP allows researchers to use libraries like HuggingFace [2] to use pretrained models by providing a private drop-in replacement for TensorFlow’s SGD.
>
> *Comment*: The author mentioned in line 258 that “…TensorFlow and PyTorch both leverage the optimized cuDNN kernel in the non-private setting for performance, but cannot in the private setting, leading to significantly worse performance.” However, the authors did not explain why cuDNN cannot be used in the private setting. Is it just an engineering issue or not feasible at all?
>
> *Response*: We claim that investing the engineering effort to create these custom kernels is not necessary, as one can achieve similar performance by simply using the XLA compiler. Furthermore, creating separate cuDNN kernels for each architecture and use-case is a significantly larger engineering effort than utilizing XLA and vectorization across arbitrary dimensions.
>
> *Comment*: The author mentioned in their contributions (line 77): ”…JAX is generally faster than Custom TFP. We examine and discuss compiled XLA assembly to explain the discrepancy.” However, such discussion is not present in the paper.
>
> *Response*: We thank the reviewer for this comment -- we examine this in l269 - l275 without going into too much depth. In Appendix A of the supplemental material, we attach code that generates low-level assembly instructions, and discuss differences in the assembly instructions between JAX and Custom-TFP.
>
> 1 - "Large-Scale Differentially Private BERT," arXiv.
>
> 2 - Wolf, Thomas, et al. "Huggingface's transformers: State-of-the-art natural language processing." arXiv preprint arXiv:1910.03771 (2019).

---

> > ### Author Response · Authors · 2021-08-24
> > **Follow-up**
> >
> > We just wanted to follow-up to see if this response adequately addresses the reviewer's questions. We'd be happy to discuss any of these points more in depth, or other details of our work.

---

> > > ### Comment · Reviewer_27Gt · 2021-08-31
> > > **Follow-up response**
> > >
> > > The author adequately addressed my question on scalability. It is nice to know that the result of this paper has been used in training BERT-Large with differential privacy. I have slightly increased my score from "6: Marginally above the acceptance threshold" to "7: Good paper, accept".

---

> > > > ### Author Response · Authors · 2021-09-01
> > > > **Thank you**
> > > >
> > > > Thank you for your time, attention, and comments on our paper, we appreciate it.

---

### Official Review · Reviewer_BcVc · 2021-07-15

**Rating:** 7
**Confidence:** 3

**Summary:**

The paper is presenting a survey of the different privacy preserving libraries available to researchers to date. It compares their runtime and memory usage on FCNN, CNN and LSTM.

The main contribution of this paper is providing a current state of the art of the different privacy preserving libraries.

**Limitations And Societal Impact:**

The limitations section is interesting and accurate as far as I can see.
One question I would have is about the claim that current privacy preserving models are small. Is that due to their increased cost and is that claim going to still be true given the improvements presented here that show that the extra cost is actually minimal.

**Main Review:**

The main high level concerns about the paper are the following:
- What is the difference between Section 3 and 4? In particular, Section 3 provide both the experimental setup as well as interpretation of the results and Section 4 seems to be rephrasing these interpretations. I think that separating Section 3 to be only the experimental setup and Section 4 being only result interpretation would make the overall paper clearer and less repetitive.
- Some of the comparisons, especially in Table 1 are a bit misleading.
  - The LSTM results discussed in particular seem to be a special case because most methods do not actually work in that context
  - While the description discusses the overhead due to moving from non-private -> private, the table put next to each other the private numbers on one hand and the non-private ones on the other hand. Making it hard to relate to the description.
  - If the goal is to show the absolute runtime while doing privacy preserving training, then the table layout is fine but the non-private section is not useful and the description should focus on that.
- The equations for CRB and BackPACK are not very useful. While the dimension size are introduced, none of the other symbols are which makes it hard to make sense of what these equations are representation and why they are important. Maybe an illustration for CRB would make the point come across more effectively, without the need for introducing all the terms.
- The evolution of the runtimes as a function of the batch size is shown in all the graphs but never discussed in details.
- l312-313, the claim than "using JAX can almost entirely remove the overhead of DPSGD" is a bit strong, especially because for LSTM, it is still significantly slower based on Table 1: 3.6 -> 8.2. Other parts of the paper like l87-88 does mention with that statement that this is only true for FCNN and CNN.

Some smaller things:
- The CNN line in Table 1 doesn't match the runtimes from Figure 2 (while the two other lines match the ones in Fig 1 and Fig 3). Could you clarify that?
- In Table 1, it is a bit surprising to put jax/TFP in the same category. My understanding of this table is to recommend a given framework that will give consistently better result. But recommending one of the two sounds much less useful as the user would have to write all his code twice.
- The sentence l88-89 that it is faster than other frameworks without privacy seems like it is comparing two things that are not comparable? why is it relevant to compare these two numbers? In particular, how do you take into account the big difference between framework in the non-private case when making that comparison?
- Some of the sentence feel a bit superlative. For example the "substancial superiority" l92. Sentences like the one l110 "the fastest approach [...] that we are aware of" seems more appropriate.
- You mention the jit compilation time l199-200. Could you give an order of magnitude for this? I have memories of some complicated architecture taking hours to compile in some older frameworks. Is that the order of magnitude we should expect here?
- Could you clarify the claim l256-257 "XLA far exceeds alternatives in the peak batch size" From the numbers in the table 3, it seems that Tensorflow 1 with XLA actually has a smaller peak batch size than Tensorflow 1 without XLA.

Some typos, small fixes
- l35: there is an extra "are"
- Table 1 caption defines FCNN and CNN but not LSTM.
- l274-275: the sentence is not english


**Time Spent Reviewing:**

3

---

> ### Author Response · Authors · 2021-08-11
> **Response to reviewer BcVc**
>
> We thank the reviewer for their detailed comments and careful reading. We appreciate the many comments and suggestions, and will address some of the questions below. For anything not addressed here, we will certainly take these into account in the final version of the paper. In particular, we will avoid overly broad claims by being more precise with our claims in the indicated areas. Feel free to ask regarding any point we do not clarify on below.
>
> *Comment*: What is the difference between Section 3 and 4? In particular, Section 3 provide both the experimental setup as well as interpretation of the results and Section 4 seems to be rephrasing these interpretations. I think that separating Section 3 to be only the experimental setup and Section 4 being only result interpretation would make the overall paper clearer and less repetitive.
>
> *Response*: Thank you for the feedback, we sought to show the results in Section 3 and explain the underlying phenomenon in Section 4. We will improve the content of these sections to reduce the repetition.
>
> *Comment*: The LSTM results discussed in particular seem to be a special case because most methods do not actually work in that context
>
> *Response*: We agree, and hoped to illustrate that the specialized kernels hindered performance in the DP use case, which is key motivation for using the XLA compiler over hand-written solutions. We will emphasize this fact to provide clarity for readers.
>
> *Comment*: The CNN line in Table 1 doesn't match the runtimes from Figure 2 (while the two other lines match the ones in Fig 1 and Fig 3). Could you clarify that?
>
> *Response*: This was a transcription mistake: the runtime in Table 1 contains the MNIST experiment from the supplementary materials, while Figure 2 contains the CIFAR 10 experiment. We thank the reviewer for pointing out this error and we will rectify it.
>
> *Comment*: The sentence l88-89 that it is faster than other frameworks without privacy seems like it is comparing two things that are not comparable? why is it relevant to compare these two numbers? In particular, how do you take into account the big difference between framework in the non-private case when making that comparison?
>
> *Response*: We initially had the comparison to demonstrate the mitigation of the privacy overhead when using JAX compared to non-private training (vanilla SGD). We agree with the reviewer that this comparison is not necessary and can be removed from the manuscript.
>
> *Comment*: You mention the jit compilation time l199-200. Could you give an order of magnitude for this? I have memories of some complicated architecture taking hours to compile in some older frameworks. Is that the order of magnitude we should expect here?
>
> *Response*: In the worst case in our experiments, the JIT compilation time took around 10x longer than the running time for an epoch. Please note that that is an upper bound and it’s likely that as the XLA compiler matures, this time will also begin to decrease.
>
> *Comment*: Could you clarify the claim l256-257 "XLA far exceeds alternatives in the peak batch size" From the numbers in the table 3, it seems that Tensorflow 1 with XLA actually has a smaller peak batch size than Tensorflow 1 without XLA.
>
> *Response*: Thank you for pointing this out. Later in the paper we explain that TensorFlow 1 does not leverage XLA as effectively as JAX or TensorFlow 2, which is why we don’t consider it truly XLA compiled (operations are simply clustered via XLA). We will clarify this sentence and emphasize this fact.
>
> *Comment*: The limitations section is interesting and accurate as far as I can see. One question I would have is about the claim that current privacy preserving models are small. Is that due to their increased cost and is that claim going to still be true given the improvements presented here that show that the extra cost is actually minimal.
>
> *Response*: Training very large models with differential privacy is often infeasible for multiple reasons. As the reviewer identifies, one reason is the increased running time, which our work partially alleviates. However, another challenge is with respect to accuracy: the magnitude of the noise injected scales proportional to the square root of the number of parameters, which means that extremely large models with billions of parameters receive very noisy gradients. That being said, our work has enabled effective and efficient large scale fine-tuning of models like BERT.[1] From a software engineering perspective, our implementation Custom-TFP allows researchers to use libraries like HuggingFace [2] to use pretrained models by providing a private drop-in replacement for TensorFlow’s SGD.
>
> 1 - "Large-Scale Differentially Private BERT." arXiv
>
> 2 - Wolf, Thomas, et al. "Huggingface's transformers: State-of-the-art natural language processing." arXiv preprint arXiv:1910.03771 (2019).

---

> > ### Author Response · Authors · 2021-08-24
> > **Follow-up**
> >
> > We just wanted to follow-up to see if this response adequately addresses the reviewer's questions. We'd be happy to discuss any of these points more in depth, or other details of our work.

---

### Official Review · Reviewer_7jTZ · 2021-07-19

**Rating:** 7
**Confidence:** 4

**Summary:**

The authors proposed methods to improve the runtime of Differential Privacy SGD using language primitives. DPSGD suffers from performance degradation compared to SGD as most of the libraries are optimized for batch computation. The authors provided implementation of a high performance DPSGD in JAX and Tensorflow. They also compared existing libraries for DPSGD and show that JAX has a superior performance.

**Limitations And Societal Impact:**

-The authors mentioned the limitations and possible societal impact of their work.

**Main Review:**

Originality:
- The authors provide new implementations for DPSGD in JAX and tf2.
- They have listed and reviewed all the previous work on DPSGD.

Quality:
- The experiment are done on small models that are used for differential privacy research on well known dataset.
- The authors also compare their implementation with other existing methods in terms of performance and memory footprint for FNN, CNN, and LSTM models.
- They also provided an ablation study on their implementations.

Clarity:
- The paper is well written and well organized and was easy to follow.

Significance:
- DP researchers and practitioners might find the work and its results interesting since it provides them with better performing libraries.


**Time Spent Reviewing:**

3

---

> ### Author Response · Authors · 2021-08-11
> **Response to reviewer 7jTZ**
>
> We thank the reviewer for their kind and encouraging words. We appreciate that they recognize the quality and importance of our work and investigation.

---

### Official Review · Reviewer_hnfP · 2021-07-31

**Rating:** 3
**Confidence:** 3

**Summary:**

The paper explores how to accelerate the Private SGD algorithm via JIT compilation and vectorization. Specifically, the author benchmarks several frameworks and libraries for DPSGD, extends TFP with vectorization and XLA compilation, and compare the runtime speed with existing libraries.

**Limitations And Societal Impact:**

The limitations and potential negative societal impact is well addressed.

**Main Review:**

*Originality: Are the tasks or methods new? Is the work a novel combination of well-known techniques? (This can be valuable!) Is it clear how this work differs from previous contributions? Is related work adequately cited?*

I would say the methods is not new. Just-in-time compilation is a well-known techniques which is adopted by deep learning frameworks in these years, like XLA, TVM, Glow, etc. Static graph optimizations and HPC techniques like vectorizations is also well adopted in frameworks.

*Quality: Is the submission technically sound? Are claims well supported (e.g., by theoretical analysis or experimental results)? Are the methods used appropriate? Is this a complete piece of work or work in progress? Are the authors careful and honest about evaluating both the strengths and weaknesses of their work?*

The submission is technically sound and benchmark result is convinced.

*Clarity: Is the submission clearly written? Is it well organized? (If not, please make constructive suggestions for improving its clarity.) Does it adequately inform the reader? (Note that a superbly written paper provides enough information for an expert reader to reproduce its results.)*


The paper can be improved by stressing the novel points and technique/methods. Too much content is focus on previous works and benchmark.

*Significance: Are the results important? Are others (researchers or practitioners) likely to use the ideas or build on them? Does the submission address a difficult task in a better way than previous work? Does it advance the state of the art in a demonstrable way? Does it provide unique data, unique conclusions about existing data, or a unique theoretical or experimental approach?*

I appreciate the work, that it can be beneficial for the community to have a fast DPSGD library. But it is more like an engineering implementation with mature techniques and is lack of novelty for me.

**Time Spent Reviewing:**

1.5

---

> ### Author Response · Authors · 2021-08-11
> **Response to reviewer hnfP**
>
> We thank the reviewer for their reading, and for their confidence in the soundness and importance of the work. We agree that a large part of our work can be considered engineering: however, a large part of modern ML research is engineering, and we believe quality work in this direction is at home in NeurIPS and neighboring conferences.
>
> Turning to the matter of novelty: we want to emphasize that, while some of our findings may be natural to experts in ML performance engineering, aspects were novel to domain experts in several areas. For instance, our improvements were not previously implemented by the TensorFlow Privacy team, which includes experts familiar with the low level workings of TF, despite slow running time of DPSGD being a significant pain point. Similarly, in correspondence with a developer of JAX, they were impressed by the magnitude of the performance improvement realized in our paper. Finally, another group recently used our techniques to enable efficient fine-tuning of BERT-Large [1]. In all cases, our findings either exhibited surprising behavior to domain experts or enabled functionality which may have previously been intractable. Thus, particularly to the privacy community which our paper targets, we believe this demonstrates both novelty and significance of our work.
>
> We will also include further details of relevant engineering challenges in the final manuscript, which include the training pipeline structure due to the nature of the vectorization.
>
> [1] "Large-Scale Differentially Private BERT" arXiv.

---

### Decision · Program_Chairs · 2021-09-27

**Decision:**

Accept (Poster)

**Comment:**

This paper proposes an approach to speed-up practical implementations of DP-SGD, a popular approach to privacy-preserving machine learning.

The reviewers recognized the significance of the work and its impact for practitioners. The author response helped to further clarify some points. One reviewer (who did not participate in the discussion) was not convinced by the novelty, as the contribution could be seen as mainly engineering. But in the end the previous arguments prevailed.

Therefore, the paper is accepted. The authors are encouraged to incorporate the additional details and discussion from the response.